# A Text-Mining and Bibliographic Analysis of the Economic Development Literature: 1959–2020

**Li Fang** 

Department of Urban and Regional Planning, Florida State University, Tallahassee, FL 32308, USA; lfang3@fsu.edu; Tel.: +1-850-644-4512

**Abstract:** In this paper, I conducted a systematic review of the economic development literature from 1959 to 2020 to reveal the ebbs and flows of major research topics, combining the text mining technique with the bibliographic analysis. Topics such as "regional development" and "sustainable development" have gained importance over the decades, while "development strategy" has lost its prominent status. New topics such as "climate change" and "developing countries" have emerged in recent years. An analysis of the citation network reveals three distinctive research trajectories: One engages with the concepts of the creative class, human capital, clusters, and art and culture. Another centers around the topic of "regional development", and a third, smaller group studies "new industrial district". This study helps researchers understand the evolution of the roadmap in the field of economic development, properly situate their own work in the literature, identify influential citations, and discover novel research topics in need of further exploration.

**Keywords:** economic development; text mining; citation network; bibliographic analysis; quantitative literature review

## 1. Introduction

Economic development has always been a key topic of interdisciplinary academic inquiry since the publication of Adam Smith's *The Wealth of Nations* in 1776 [1]. Why some regions are rich while others are poor is among one of the most important puzzles of human history and closely related to millions of people's livelihood. As Lucas (1998) said, "the consequences for human welfare involved in questions like these are simply staggering: Once one starts to think about them, it is hard to think about anything else" [2]. As a result, over the past a hundred years, researchers have explored various questions related to economic development, and examined the determinants, process, outcomes of development and policy tools to stimulate economic prosperity [3–5]. As a result, a researcher in this field, while enjoying the richness of the relevant literature, also bears the costs of constantly getting lost in the deep weeds of the literature without the ability to see a clear theoretical and methodological thread.

Some qualitative reviews have provided some guidance by summarizing over studies on a specific subtopic. They have explored various determinants of economic development, such as market [6], institutional arrangement [7–9], infrastructure [10,11], education and research institutions [12,13], technology [14,15], and more recently, entrepreneurship and social networks [16]. Some examine the development process [17], the economic, social, and environmental outcomes [18–22], and the evolution of policies and practices [4,23]. A few books go beyond a specific subtopic and are more comprehensive in coverage. For example, Aghion and Durlauf (2005) package theories, empirics, and policies in the *Handbook of Economic Growth* [3], and Reinert, Ghosh, and Kattel (2016) complement the mainstream discussions with alternative theories, with focuses on historic, geographical, and other perspectives [5].

Meanwhile, the advancement of machine learning technology and bibliographic analysis has made it easier to efficiently summarize a large number of texts, reveal research

topics and trajectory, and identify citation relationships. Researchers in social sciences have already adopted such technology to understand research trajectories. For example, Cruz and Teixeira (2010) have adopted a bibliometric analysis to quantify the evolution of the cluster literature from 1962 to 2008, which precisely maps out the dynamics of this group of literature in regional science [24]. Moreover, Gobster (2014) has adopted text mining technology to examine the evolution of research topics among the publications of *Landscape and Urban Planning* in the past 40 years [25]. Similarly, Kaplan and Mapes (2015) examined topics of geography dissertations in the past 120 years with text mining analysis. More recently, Fang and Ewing (2020), using a topic modeling method, revealed major research themes in three urban planning journals in the past 30 years [26]. Similar methods can be applied to the study of the economic development literature to quantify research trends.

This paper takes on the task of quantifying the research dynamics in the economic development literature to complement the existing qualitative reviews. I collected the title, abstract, and keywords from 6464 published articles and book chapters on the topic of "economic development" in the fields of Urban Studies and Regional and Urban Planning from 1959 to 2020, and combined text mining technique and bibliographic analysis to identity and track the changes in research topics and citation dynamics. The text mining analysis reveals the ebbs and flows in the field, enhancing researchers' abilities to grasp the broader research trajectory, and thus better situate their work in the literature. I've also highlighted emerging topics in the most recent five years, which may continue to be the major focus in the coming years and could potentially inspire future studies and facilitate the organization of academic communities with shared interests in these new directions. The bibliographic analysis reveals citation networks among articles, journals (disciplines), institutions, and countries, which shows how researchers, disciplines and institutions interact in the field of economic development and identifies structural holes in the network that can become future breakthroughs.

The paper is organized as follows. The next section introduces the concept of "economic development", setting the stage for the following analysis. Section 3 introduces the collection of the literature and the analytical methods. Section 4 presents and discusses the results from the text mining and the bibliographic analysis, and Section 5 concludes with main findings and implications for future research.

## 2. The Concept of Economic Development

The examination of what causes economic development started as early as Adam Smith's *The Wealth of Nations* in 1776 [1], when principles of growing a nation's wealth emerged. With the rapid development of mathematical tools, economists accumulated a whole collection of literature on growth theory and growth models [2,27]. In these models, production factors such as capital, land, labor, technological advancement, human capital, and institutions have been found to be fundamental in supporting a nation or region's economic prosperity.

However, criticisms about the narrow focus on growth have also arisen, pointing out its ignorance of social and economic inequality, environmental quality and sustainability, and human happiness and quality of life. As a result, scholars started to explore alternative definition for economic development that goes beyond growth and emphasizes other aspects of development. For example, Blakely and Leigh (2013) develops a three-part definition: "First, economic development establishes a minimum standard of living for all and increases the standard over time. Second, economic development reduces inequality. Third, economic development promotes and encourages sustainable resource use and production" [28]. That three-layer definition has greatly enriched the encompass of the concept of economic development and redirected it towards an equity- and sustainability-oriented approach.

The practice of economic development has experienced a similar evolution. For example, Bradshaw and Blakely (1999) summarized three waves of economic development strategies, which shifted from a traditionally incentive-based approach from a regional

collaborative and cluster-oriented approach [4]. Recently, an equity- and sustainability-oriented practice has also been found to be on the rise [29]. Overall, the concept of economic development has evolved, both in theory and practice, from a growth-focused approach towards a more diversified and complex concept, and in recent years, a more inclusivity- and sustainability-oriented approach.

## 3. Materials and Methods

### 3.1. Materials

For the purpose of understanding the research landscape on economic development, I need to collect the extant literature. I choose to conduct my research for the literature through the Web of Science Core Collection (Beijing, China). The Web of Science is one of the world's most trusted publisher-independent global citation database; it provides a multidisciplinary and comprehensive platform that allows one to track ideas across disciplines and time from almost 1.9 billion cited references from over 171 million records [30]. Admittedly, using only the Web of Science Core Collection as the data source leads to an insufficiently representative sample. Nonetheless, the Web of Science Core Collection does contain a large number of publications and provide a readily usable format of the data for the analyses of this study. Moreover, no publication database can truly deliver a representative sample.

Then, in order to understand the dynamics of research topics and the citation network among articles, journals, institutions, and countries. I've used CiteSpace (Philadelphia, PA, USA), a Java application designed for visualizing and analyzing trends and patterns in scientific literature, to conduct these analyses. CiteSpace is a powerful tool developed specifically for the purpose of conducting such analyses, which is a perfect fit for the purpose of this study. Figure 1 illustrates the steps I go through to conduct these analyses.

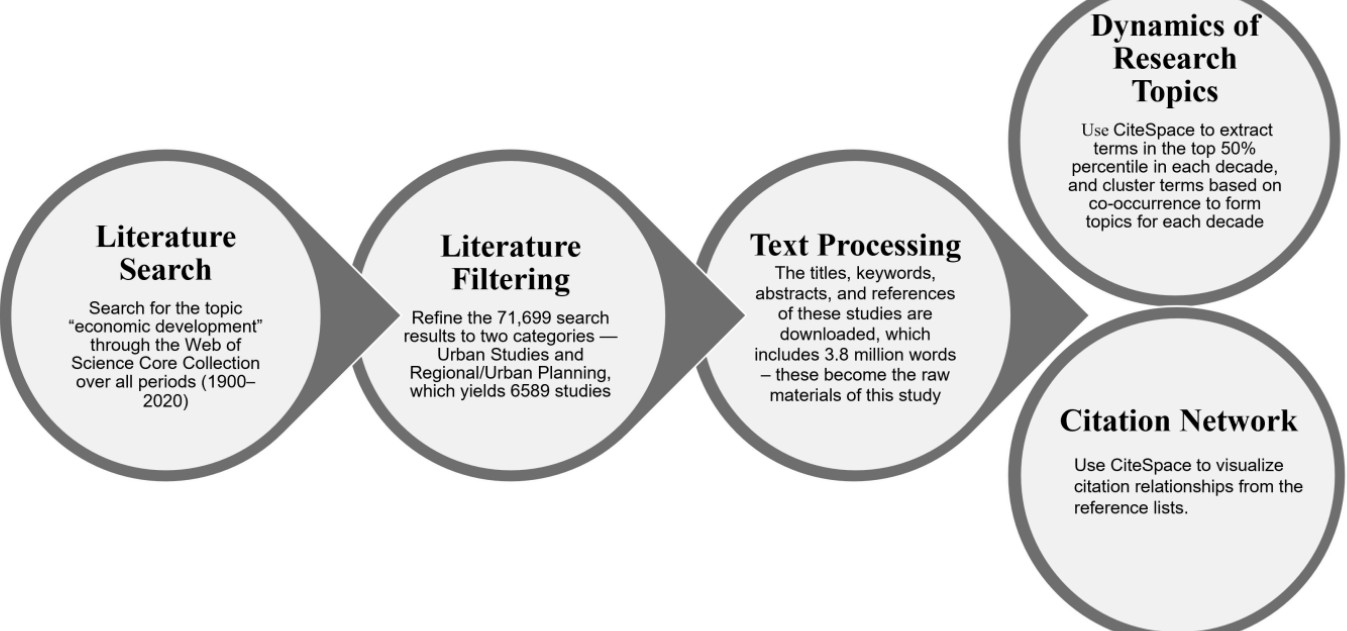

**Figure 1.** The methodological procedure.

### 3.2. Methods

Data are collected through the following procedure, which is similar to the standard practice in the literature [24]. I searched for the topic "economic development" in the Web of Science Core Collection over all periods (1900–2020). This search returned a collection of 71,699 studies, whose title, keywords, or abstracts is related to economic development. The earliest publication date of these studies is 1959. However, these studies are from

a massive variety of different fields, making the review incoherent. Thus, I further refined the categories of studies to those that are in the urban or regional contexts, which include two categories in the Web of Science database—Urban Studies (2892 studies) and Regional/Urban Planning (4764 studies). This is sensible as the field of Urban Studies includes studies on urban economic development, while the field Regional/Urban Planning includes regional development as well as economic development as a one of the major subfields of urban planning. This yielded a total of 6589 studies because one article can fall into multiple categories. These studies and their titles, keywords, abstracts, and references become the raw materials for this paper's bibliographic analysis, which includes 3.8 million words. The large number of texts in this collection makes it more suitable for a text mining analysis than a human read.

The number of economic development publications has increased drastically over time (Figure 2). The surging number of publications makes a systematic review more useful for researchers to effectively grasp what has been studied and how the research topics have evolved. These articles come from a variety of journal outlets, including *Regional Studies*, *Urban Studies*, *Economic Development Quarterly*, *European Planning Studies*, *Growth and Change*, *Journal of Developing Areas*, *Cities*, and *Urban Affairs Review*.

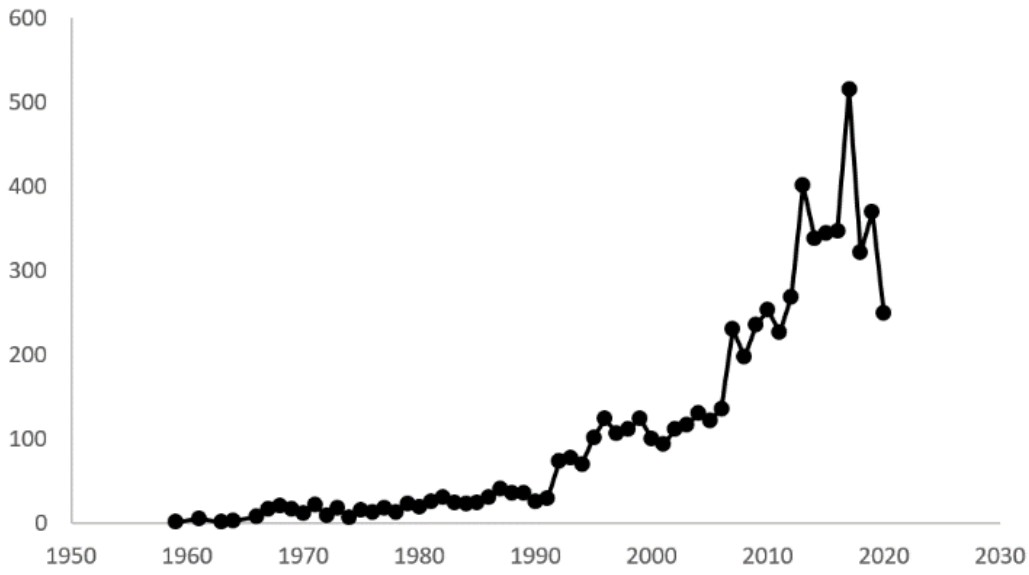

**Figure 2.** Number of economic development publications over time.

I've focused the analysis on two issues: the dynamics of main research topics and the citation network among articles, journals, institutions, and countries. I first explored the clusters of terms in the collected studies by decades to reveal the ebbs and flows of research topics. CiteSpace extracts terms from texts of the collected article titles, keywords, and abstracts. I included terms whose occurrences are in the top 50% percentile in each decade to focus the analysis on important terms and make the visualization digestible. Terms are clustered according to their frequency of appearing in the same article. I then named each cluster, i.e., topic, based on the most prominent terms. In addition, CiteSpace allows an examination into the evolution of each topic—revealing how key terms have changed within one topic. Through these analyses, a research trajectory is revealed showing the main research topics and their evolution.

I then focused on the citation network. CiteSpace takes the reference lists from the collected articles and analyzes the citation relationships. A citation relationship between two articles is visualized as a connection in a network. I've included articles, journals, institutions, and nations (as nodes in the citation network) whose citations are among

the top 50% percentile in at least one decade. This choice is made to focus on the most influential nodes rather than nodes with sporadic citation links. Through this analysis, I am able to track the most influential papers and the group of papers that cite them. I also explored the overall citation structure of the field—whether most articles cite a few prominent papers or distinctive groups cite very different papers.

I also examined the citation networks among journals, institutions, and countries. These exercises help reveal whether there is a hierarchical citation structure among journals and help identify knowledge gaps. This exercise can also inform us about whether economic development is truly an interdisciplinary field with journals cite across disciplinary boundaries. The analyses of institutions and countries can show the geographical landscape of the economic development research and help us understand whether citation, as a trail of knowledge spillover, is significantly affected by geographical proximity.

While the methods adopted in this study are certainly useful in revealing important patterns and insights, they inevitably suffer from a few limitations. First, the collection of the literature is incomplete and limited by the Web of Science Core Collection and Web of Science's classification of the literature. Nonetheless, as mentioned above, the Web of Science database remains one of the most comprehensive. Second, depending on different authors' writing styles, important terms may be left out of article titles, abstracts, and keywords. Moreover, since I focus on terms whose occurrence is among the top 50% percentile, I leave out potentially important but less frequently occurring terms. This could bias against new and emerging terms. To mitigate this concern, I specifically conduct an analysis on publications in the recent five years, to identify newer and emerging trends. Third, while efficient, text mining is also a crude way of summarizing literature. Compared to more in-depth qualitative reviews, it trades off details and depth for breath and broader coverage.

## 4. Results

### *4.1. The Dynamics of Research Topics*

Nine main topics emerge among the literature, as shown in the first column of Table 1. Topics are ranked by the percentage of texts they account for, which can be interpreted as their prominence. The most prominent topic is "driving forces of development", accounting for 31% of the texts. The secondary topic "production factors" accounts for 30% of the texts. "Local development" is the third most important topic (10% of the texts), focusing on development issues at a local scale. Following it are "central city development" and "developing countries", which account for 7% and 6% of the texts, respectively. The former zooms in to develop in the central city, while the latter zooms out and looks at development from a global south perspective. Then "regional study" and "small businesses" each account for 5% of the texts. Finally, "innovation" and "policy and plan" are the least important topics, each accounting for 3% of the texts.

The focus of each topic has also evolved, as shown by the dynamics of keywords in Table 1. For example, the most prominent topic "driving forces of development" has shifted from a global-focused market-driven vs. government-driven approaches (featured by the keywords forced labor and Soviet industrialization in the 1960s) to a more local-focused exploration of development endeavors (featured by keywords such as metropolitan area, economic activity, and economic development incentives after 2000).

Main topics also vary by decade, as shown in Table 2. The overall landscape of the economic development literature has become more diverse, covering many different topics after 1990, compared to the 1970s and 1980s. Specifically, only one topic exists in the 1970s and 1980s, but for each decade after 1990, there are at least six topics.

**Table 1.** Topic and topic keywords dynamics.

| Topic (Percentage of Texts)/Decade | 1960s | 1970s | 1980s | 1990s | 2000s | 2010s |
|---|---|---|---|---|---|---|
| Driving forces of development (31%) | forced labor, Soviet industrialization, high-level manpower | foreign assistance programs, local currencies, financial sector | | | metropolitan area, economic activity, economic development incentives | economic development incentives, value capture, great recession |
| Production factors (30%) | | | public policy, socio-economic development, metropolitan area | investments, sports, pension | small towns, business | human capital, socio-economic development, social capital |
| Local development (10%) | | | high technology, local entrepreneurship | local politics, urban regeneration, business coalitions | local governments, social capital, human capital | human resources, urban regeneration, local authorities |
| Central city development (7%) | | central city, minority opportunity | new role, relationships | | | |
| Developing countries (6%) | | | negligence, decline, developing countries | location analysis, world bank, public goods | central Europe, foreign direct investment | gross domestic product, foreign direct investment, developing countries |
| Regional study (5%) | | perceptual dimension | | technological change, development process | sustainable development, regional study, social development | European union, knowledge economy, regional innovation system |
| Small businesses (5%) | | rising energy prices | small firms | | | |
| Innovation (3%) | | | urban systems, innovation, interrelationships | global economy, innovation process, local cluster | industry cluster, developed countries, innovative activity | |
| Policy and plan (3%) | | | economic development policies, evaluation study | | case study, rural economic development | |

**Table 2.** Main topics over time.

| Topic Rank/Year | 1959–1970 | 1971–1980 | 1981–1990 | 1991–2000 | 2001–2010 | 2011–2015 | 2016–2020 |
|---|---|---|---|---|---|---|---|
| 1 | driving forces of development (58% *) | production factors (100%) | local development (100%) | environmental management (14%) | local development (28%) | culture industry and creative class (20%) | sustainable development (20%) |
| 2 | migration (8%) | | | local development (13%) | regional study (18%) | production factors (19%) | developing countries (18%) |
| 3 | production factors (8%) | | | production factors (13%) | innovation (18%) | regional study (17%) | local development (17%) |
| 4 | industrial location (8%) | | | regional study (12%) | policy and plan (15%) | local development (16%) | regional study (16%) |
| 5 | Soviet industrialization (8%) | | | developing countries (11%) | driving forces of development (14%) | sustainable development (15%) | production factors (16%) |
| 6 | political analysis (8%) | | | community development (11%) | sustainable development (6%) | rural development (14%) | climate change (8%) |
| 7 | | | | sustainable development (9%) | | | driving forces of development (5%) |
| 8 | | | | innovation (9%) | | | |
| 9 | | | | policy and plan (8%) | | | |

* The percentage of texts a topic accounts for is presented in parentheses.

Some topics have remained important throughout the study period, such as "production factors" and "local development", which appear among the top five topics across

multiple decades. In contrast, various topics have experienced significant changes in status. For example, "driving forces of development", while remaining an important topic throughout most of the study period, its importance has eroded over time: It starts as a top topic in 1959–1970 but falls into the least important in the most recent period (2016–2020). Still, some topics have surged over time. These include "regional study", which emerges as scholars and practitioners started to take a regional collaborative approach to manage economic development in the 1990s [4,31] and remains an important topic thereafter. Another example is "sustainable development", which also emerges in the 1990s and gains importance over time; it becomes the most important topic in the recent five years, likely due to the socio-economic issues brought by climate change and social inequality. This signals increasing interests in striking a balance between economic growth, environmental quality, and social equity [32].

Seven topics feature the most recent five years. "Sustainable development" has become the top priority, accounting for 20% of the texts. Meanwhile, "climate change" has emerged as a separate topic, accounting for 8% of the texts. These findings show that compared to previous periods, the research focus now is placed more on social justice and environmental quality. Another topic that has gained importance is "developing countries". This topic accounts for 18% of the texts and ranks second among all topics. This may be due to an increasing interest in the developing world, or a decentralization in research, with more authors from developing countries publishing their articles studying their nations. These diverse perspectives and experiences in developing countries provide rich contexts for us to deepen our understanding of development issues. The various settings also provide a great testing ground for the generalizability of economic development theories and policies, which are developed mostly based on the contexts of developed nations. Other prominent topics include "local development", which accounts for 17% of the texts, "regional study" and "production factors", each account for 16%, and "driving forces of development", accounting for 5%. The evolution of these topics also highlights the "spatial" element of the economic development issues; the concept of economic development has evolved from a more abstract and national-level notion to a more concrete, space-based, and multi-spatial-scaled concept with a clear spatial scope specified—be it regional level, city level, or within-city level.

*4.2. The Citation Network of Papers*

The citation network of the literature clusters into three major groups, as visualized in Figure 3 in different colors. Two features stand out. First, the three groups of papers are quite distinctive from each other, with few cross-group citations. Thus, a typical researcher tends to choose one stream of the literature and place his or her own work inside that group. Seeking potential integration and interactions across groups may set off a promising future research direction. Second, while the largest group—"creative class, art and culture"—has some studies cited significantly more than others, the other two groups are quite decentralized in terms of citations. Overall, the academic landscape is not dominated by a few highly influential studies.

Zooming into the three groups of research, the largest group, "creative class, art and culture", centers around the Florida book *The Rise of the Creative Class* [33]. This book develops the concept of "the creative class", and emphasizes diversity and creativity as the main drivers for economic prosperity for modern cities. This book is the most heavily cited among this group, and shortly after its publication, a few related articles also published and become heavily cited. These include Peck (2005) [34] and Markusen (2006) [35], which critically engage with the concept of the creative class; building on Florida (2002) [33], these authors have provided empirical evidence and raises concerns about this concept. Markusen and Schrock (2006) focus specifically on one type of creativity and examines the relationship between art and culture and economic development [36]. This specific focus on art and culture clusters has later set off a wave of related studies [37,38]. Most papers in this group were published during the period 1998 to 2010 and most influential authors are

based in North America. As mentioned above, this group of literature shows some degree of concentration in citation pattern, with these studies named above receive many more citations than the rest.

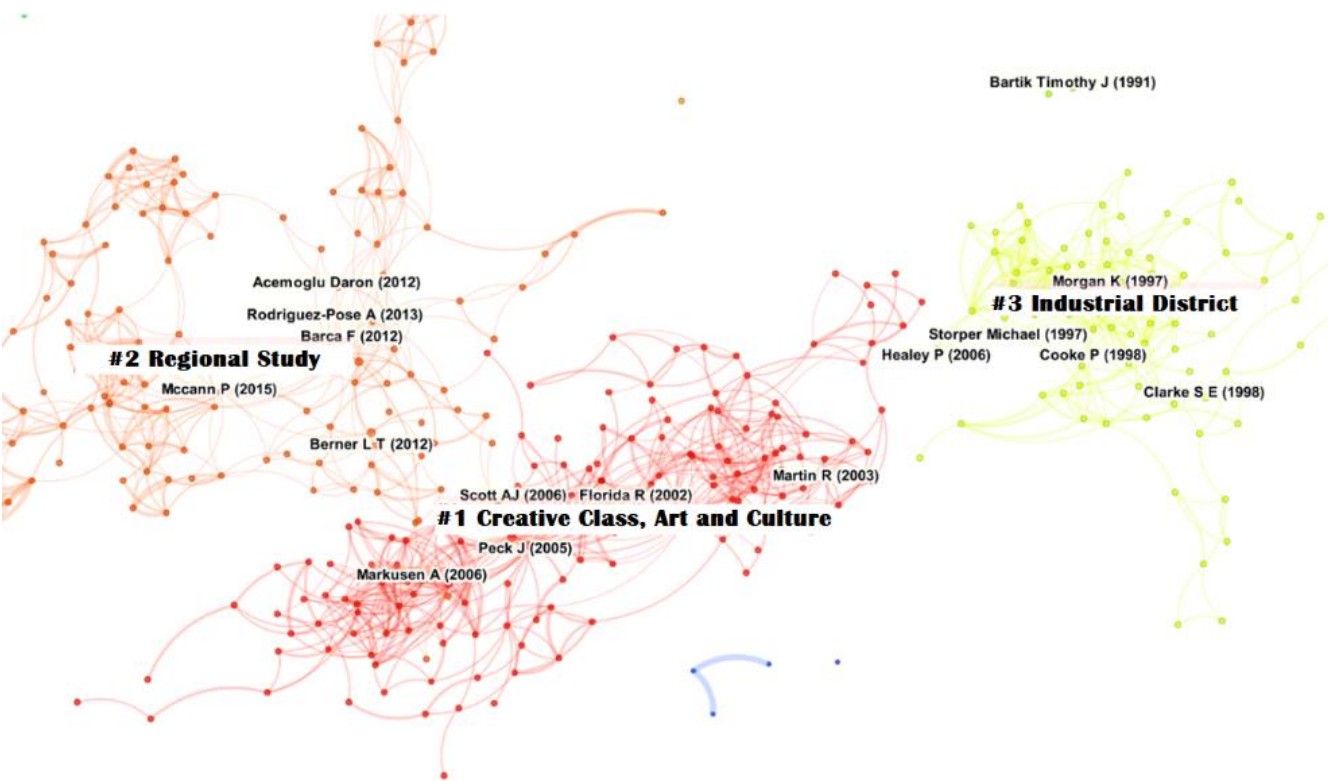

**Figure 3.** Three major citation clusters.

The second largest group—"regional study"—has four most-frequently-cited studies. Barca, Mccann, and Rodríguez-Pose (2012) compares different policy interventions to promote regional development [39]. More specifically, Rodríguez-Pose (2013) engages with the importance of institutions in regional development [40], and McCann and Ortega-Argilés (2015) discusses smart specialization in the context of the European Union [41]. More recently, from a relational perspective, Spigel (2017) examines the entrepreneurial ecosystem within regions [42]. These studies mostly are published in regional science journals such as *Regional Studies* and *Journal of Regional Science*. Most influential authors are based in Europe, especially the UK, fueled by the practice of the European Union that promotes regional collaboration and integration. Moreover, this group of literature is the most recent among all three clusters, with the majority of studies published during the period 2007 to 2020. This finding again echoes the previous conclusion that the spatial scale of economic development studies become more confined in more recent studies. Also, despite having four most frequently cited studies, this group of studies in general forms a decentralized citation pattern, i.e., many other studies also receive a significant number of citations.

The third largest group—"industrial district"—is the most decentralized in citation pattern with many studies cited plenty of times. Among the most cited studies, Morgan (1997) takes the Welsh Development Agency as an example to examine the shift in development approaches, from the provision of hard infrastructure to promoting innovation, and develop business services, skills and social capital [43]. Porter (1998) sets off a wave of studies examining industrial districts and how they improve the competitiveness of a region [44]. Journals that publish these articles come from a variety of disciplines—urban studies, regional science, urban planning, and business.

To reveal emerging citation clusters, I also focused my analysis on articles published in the most recent five years. Five citation clusters emerge. Rodríguez-Pose (2013), Barca

et al. (2012) and McCann and Ortega-Argilés (2015) lead a cluster titled "smart special-ization" [39–41]. This is an emerging group of studies discussing how to properly target regional specialization for gaining global comparative competitiveness. Its emergence is also directly related to the promotion of "smart specialization strategies" in the European Union to adopt a combination of industrial, educational and innovation policies to boost regions' comparative advantages in a few carefully selected priority areas.

Wolf-Powers et al. (2017) connects a group of studies on "industrial districts" [45]. This group of studies has taken a detailed look into the theories and practices of "industrial districts" across the world. Specifically, Wolf-Powers et al. (2017) examined the recent trends of the "maker movement" in the United States, featuring small-scale manufacturing development that values creativity and craftsmanship [45]. Compared to the traditional large-scale "industrial districts", this type of districts attracts makers who are more entrepreneurial and creative. A shift in focus from large-scale manufacturing or high-tech industrial districts to smaller-scale entrepreneurial districts also reflects a change in local economic development practice that focuses more on local entrepreneurs and small businesses [4,46].

Wu (2016) inflames a wave of studies on "spatial analysis", which examines the spatial structure of economic development [47]. Specifically, Wu (2016) discussed the emerging trend in China to embrace regional coordinated spatial planning to harness the power of clusters [47]. Kline and Moretti (2014) feature a group of studies on "agglomeration economies", aiming at unleashing the power of urban agglomerations—concentrations of residents, firms, and workers—to promote economic development [48]. Finally, Smith and Bagchi-Sen (2012), Huggins, Johnston, and Stride (2012), and Audretsch (2014) center a cluster focusing on the topic "university and knowledge" [49–51]. They study the role of universities in generating and commercializing new knowledge.

These five emerging clusters may continue to grow and branch out in the near future. An increasing enthusiasm on smart specialization, entrepreneurship and developing countries is revealed, which is closely related to recent economic development practices and therefore forms a fertile ground for researchers to explore. The network structure among these five clusters also reveals that while the clusters "smart specialization", "industrial districts", and "agglomeration economies" are closely connected with each other through citations, the other two clusters—"spatial analysis" and "university and knowledge"—are currently disconnected from the other topics. The intersections between the latter two groups of studies and the former three groups may be promising areas for further exploration. For example, topics such as how universities function as a part of an entrepreneurial industrial district may be worthy of further probe.

### 4.3. The Citation Network of Journals

Many journals publish economic development research, but a few distinguish themselves as the most frequently cited. Among these journals, three major clusters surface. The largest cluster includes mainly urban studies, urban planning, and regional science journals—*Urban Studies*, *Regional Studies*, *Economic Development Quarterly*, and *Environment and Planning A* for example. These journals frequently cite each other and form a tightly connected knowledge cluster, though they do not strictly come from the same discipline. This demonstrates the truly interdisciplinary nature of this field.

A second cluster is formed by economic journals, such as *American Economic Review* and *Journal of Political Economy*. These journals tend to more frequently cite other economic journals, though they are still connected with the "urban and regional studies journals" cluster. This finding points to a potential gap in the knowledge spillover between the economists and other economic development researchers; more communication and collaboration ought to be encouraged.

Similarly, a third cluster, forming around the *International Journal of Urban and Regional Research* and *Urban Affairs Review*, is also less connected with the other clusters. The former publishes mainly international studies, while the latter originates from the field of sociology. The topics that these journals publish, e.g., international development and social issues,

appear to be on the rise in the territory of economic development (attested by findings in Section 4.1). However, so far, these journals have not been well integrated into the overall landscape of economic development research. Incorporating international perspectives and sociology studies could be another interdisciplinary undertaking for economic development researchers in the near future.

*4.4. The Citation Landscape: Institutions and Countries*

Finally, I've also explored the geographical citation landscape in the form of cross-citations among institutions and countries. I find three prominent institutional cross-citation clusters, each consisting of more than 25 institutional members. The most cited cluster includes mainly US universities, and among them, the University of Illinois and the University of North Carolina are the most frequently cited. Three additional universities are also cited quite often, including Florida State University, Cleveland State University, and Cornell University. This cluster studies a variety of topics, including local economic development policy, transportation expenditures, location affordability, and public art. The second cluster includes primarily European universities, and among them the most cited are Cardiff University and the University College London. This cluster focuses on the industrial crisis and economic restructuring. A third cluster is formed among mainly Asian universities or research institutions, such as the Chinese Academy of Science and the University of Hong Kong. This cluster's focus is on housing price and spatial dynamics. These three clusters are primarily formed based on geography, signaling the importance of physical distance in the production and distribution of knowledge.

I've also explored the citation pattern among nations. Five country-level clusters emerge. The largest cluster consists of 26 countries and centers around two most cited countries—the USA and the Netherlands. Their main research topic is "sustainable urban development". The second-largest cluster, consisting of 20 countries, centers around four frequently cited countries—China, the Czech Republic, Poland, and Russia. The focus of this cluster is "the knowledge-based economy" and "low-income countries". The third-largest cluster has 17 members, with Italy, Spain, and France being the most frequently cited. They focus on "technology policies" and "the transportation system". The following two clusters have the same size, each with 16 members. One centers around Australia, Germany, and South Africa, with the research focus placed on "property rights"; the other centers around UK and Canada, with the research topics focusing on "international perspectives" and "urban regeneration". The clusters of countries are less constrained by geography, compared to those of institutions. Rather, these clusters are formed based on shared interests in addressing similar pressing issues these countries face.

## 5. Discussion

This paper analyzes the main topics of the economic development literature and their dynamics from 1959 to 2020. Nine topics are identified, including "driving forces of development", "production factors", "local development", "central city development", "developing countries", "regional study", "small businesses", "innovation", and "policy and plan". Moreover, research topics experience ebbs and flows. Some topics have gained importance over time, such as "regional study" and "sustainable development". Others have lost their prominent status such as "driving forces of development". In the most recent period, 2016 to 2020, "sustainable development", "climate change", and "developing countries" have gained significant importance, signaling emerging trends towards these directions.

The citation networks among articles, journals, and institutions, and countries have also shown interesting patterns. Three major citation clusters among papers emerge. The largest citation cluster titled "creative class, art and culture" centers around Florida (2002) [33], following by a second-largest cluster titled "regional study", with McCann and Ortega-Argilés (2015) [41], Barca et al. (2012) [39], Rodríguez-Pose (2013) [40] and Spigel (2017) [42] being the most cited articles. A third citation cluster forms around Morgan (1997) [43] and Porter (1998) [44] to engage with the topic "industrial district".

At the same time, most journals that publish economic development studies frequently cite each other, forming a tightly connected citation cluster. Examples of these journals include *Urban Studies*, *Regional Studies*, *Economic Development Quarterly*, and *Environment and Planning A*. *American Economic Review* and *Journal of Political Economy* form a separate citation cluster, with a few other economic journals. *International Journal of Urban and Regional Research* and *Urban Affairs Review* also form their own citation cluster. Institutions appear to cluster based on geography. Universities geographically close to each other are more likely to cite each other and study related topics. In contrast, countries do not cluster based on geography. Instead, they cluster due to the similarity of topics, mostly related to the social issues that a country faces.

These findings pertain to both researchers and practitioners. For researchers whose interests fall into one of the major topics, this article identifies key resources for them. For each major topic, the most influential articles and those that cite them are identified so that researchers can easily trace their theoretical roots and properly situate their work in a strand of literature. Moreover, this paper also examines the dynamics of each topic, helping researchers to grasp the historical evolution of their topic and add a time horizon to their knowledge structure.

For researchers whose interests overlap with several topics, this article identifies the specific connections between these topics with citation links. In particular, if such linkages are few, then the intersection may be an area where significant contributions can be made. The researchers' work could play a bridging role in the knowledge network.

For researchers and students looking for novel topics to study, journals to submit their articles, and institutions to work in, this article also provides direct guidance. The emerging topics in the most recent five years, as well as where connections are missing among topics, i.e., structural holes in the citation network [52], could be promising fields to work on. Journals that publish a lot of economic development studies may be good places to land their articles. Besides, institutions that focus on specific topics which coincide with a researcher's interests can be destinations where they seek for jobs, visiting positions, or collaboration opportunities.

This study is also useful for practitioners to quickly understand the recent development of theories and empirical studies. Such knowledge can help them become reflective practitioners who effectively relate their professional experiences to theoretical frameworks, consciously observe their behaviors, and challenge the status quo. Moreover, a grasp of the recent development in the academic endeavor can also help practitioners identify missing links between theory and practice [53], upon which practitioners can take an initiative to address. They may start a collaboration with researchers to fill the gaps, which brings research and practice together and forward. I wish to see that happen in the near future.

**Funding:** This research received no external funding.

**Data Availability Statement:** Data for this study can be found through searching in the Web of Science database.

**Conflicts of Interest:** The author declares no conflict of interest.

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
