# Peer review of "A Text-Mining and Bibliographic Analysis of the Economic Development Literature: 1959–2020"

_urbansci, doi:10.3390/urbansci6040080_

Round 1
Reviewer 1 Report
I believe the paper is interesting, well written and easy to read.
At the same time, I think the article suffers from some issues.
Below I present more in detail my comments:
1- There is an inconsistency between the title and the content of the paper: the title (as well as many sentences throughout the texts) specifically mentions “Urban” Economic Development. However, the whole analysis does not seem to be performed on Urban Studies but rather on regional or even national development.
2- I believe the positioning of the paper itself is not convincing. It would be better, in my opinion, to discuss it in the context of text mining and bibliographic analysis instead of comparing it to qualitative and more in-depth analysis of regional economic development. Furthermore, I think the critique to the cited qualitative reviews is excessively severe.
3- The methodology used in the paper should be better explained and better contextualized. Is there a specific literature on it? Are there specific guidelines/procedures/benchmarks? Which are the limitations of this methodology?
4- I think that the choice of including “Planning” studies should be better discussed. Why including the planning literature if the main focus is regional/urban? economic development?
Overall, apart from an overview of macro-themes and macro-trends I believe the discussion should be strengthen and the value added of this paper should be better clarified and substantiated.
Reviewer 2 Report
Work well done, well presented and, in its panoramic role, enlightening. My only objection has to do with the "urban" attribute added to the concept of "economic development", beyond the link that derives from the chosen field of analysis (selected journals). The spatial factor (the city, the region, the urban... their planning) is to a certain extent absent, although it appears in the research topics... and flows into sustainable development. This is evident further on: isn't it singular that in the clusters all the themes are related with space or urban governance? Creative class, regional institutions and ecosystems, industrial district, spatial analysis... Could it be affirmed, based on the topics present in the proposed citation clusters, that there is an evolution from a more abstract concept of economic development to a more concrete one, more space based? About the usefulness of the work for researchers and practitioners I have no doubt.
Round 2
Reviewer 1 Report
Thank you for the answers to the comments and the revision.